# PARROT is a flexible recurrent neural network framework for analysis of large protein datasets

**Daniel Griffith[1,2], Alex S Holehouse[1,2]\***

[1]Department of Biochemistry and Molecular Biophysics, Washington University School of Medicine, St Louis, United States; [2]Center for Science and Engineering Living Systems, Washington University, St Louis, United States

**Abstract** The rise of high-throughput experiments has transformed how scientists approach biological questions. The ubiquity of large-scale assays that can test thousands of samples in a day has necessitated the development of new computational approaches to interpret this data. Among these tools, machine learning approaches are increasingly being utilized due to their ability to infer complex nonlinear patterns from high-dimensional data. Despite their effectiveness, machine learning (and in particular deep learning) approaches are not always accessible or easy to implement for those with limited computational expertise. Here we present PARROT, a general framework for training and applying deep learning-based predictors on large protein datasets. Using an internal recurrent neural network architecture, PARROT is capable of tackling both classification and regression tasks while only requiring raw protein sequences as input. We showcase the potential uses of PARROT on three diverse machine learning tasks: predicting phosphorylation sites, predicting transcriptional activation function of peptides generated by high-throughput reporter assays, and predicting the fibrillization propensity of amyloid beta with data generated by deep mutational scanning. Through these examples, we demonstrate that PARROT is easy to use, performs comparably to state-of-the-art computational tools, and is applicable for a wide array of biological problems.

**\*For correspondence:**
alex.holehouse@wustl.edu

## Introduction

The past decade has seen an exponential increase in the rate at which biological data is generated (*Marx, 2013*). Technological advances coupled with the falling costs of DNA synthesis and sequencing have made conducting high-throughput experiments accessible to most research labs (*Hughes and Ellington, 2017*). The affordability of being able to sequence massive quantities of DNA is transforming how molecular biologists approach research. Protein functional assays and screens are seeing increasing library sizes, which allows researchers to investigate many different sequences and variants in a single experiment. In recently published studies, it is not uncommon to find deep mutational scanning (DMS) experiments that achieve nearly complete sequence coverage or assays that test tens of thousands of peptides (*Arnold et al., 2018*; *Bolognesi et al., 2019*; *Erijman et al., 2020*; *Jones et al., 2020*; *Livesey and Marsh, 2020*; *Seuma et al., 2021*; *Sanborn et al., 2021*; *Schmiedel and Lehner, 2019*). This abundance of data being generated has the potential to answer important biological questions; however, at the same time, it also significantly complicates experimental analysis.

Coinciding with the explosion of high-throughput omics experiments has been the development of computational methods for analyzing the resulting high-dimensional biological data. In particular, machine learning approaches have emerged as popular strategies in a wide range of biological applications (*Xu and Jackson, 2019*; *Eraslan, 2019*; *Moses, 2017*). In general, machine learning approaches are effective at identifying patterns in complex datasets and extrapolating these learned

patterns to make predictions on previously untested samples. Deep learning approaches, as opposed to 'shallow' machine learning approaches, such as logistic regression, are particularly well-suited for biological applications as they can implicitly capture relevant features in order to model complex, nonlinear, biological relationships (*Min et al., 2017*; *Raimondi et al., 2019*; *Xu et al., 2020*). In the context of protein datasets, deep learning approaches offer the attractive quality of allowing researchers to simply input raw protein sequences into the model, rather than requiring an intermediate step where proteins are reduced into simplified representations (e.g., amino acid content or biophysical properties; *Raimondi et al., 2019*).

However, despite their advantages over simpler models, deep learning approaches are still a relatively specialized form of data analysis. As a result, in many domains of biological sciences, there remains a technical and conceptual barrier for labs to apply deep learning approaches to their data. In some cases, this could be reasonably attributed to preference for more interpretable simple models, rather than more accurate, but often cryptic, deep learning models (*Rudin, 2019*; *Murdoch et al., 2019*). In other cases, this lack of adoption could be due to a general unfamiliarity and inexperience with deep learning. Indeed, the field of deep learning can appear daunting for those without extensive computational backgrounds. For an untrained scientist with amenable high-throughput datasets, it may be infeasible or too time-consuming to implement deep learning models into an analysis workflow.

Here, we aim to make cutting-edge deep learning accessible to a broad audience of biological researchers through our package PARROT (Protein Analysis using RecuRrent neural networks On Training data). PARROT is designed to be a general framework for training machine learning networks on large protein datasets, then using the trained network to make predictions on new protein sequences. The user side of PARROT is an easy-to-use command line tool that is flexible enough to handle a variety of data formats and machine learning tasks. In its implementation, PARROT carries out the computational heavy lifting through implementation of a recurrent neural network (RNN). RNNs are a class of deep learning architecture originally designed for language processing applications, but have since been employed with remarkable success in biology (*Rumelhart et al., 1986*; *Lipton et al., 2021*; *Hanson et al., 2017*; *Heffernan et al., 2017*; *Almagro Armenteros et al., 2017*; *Li et al., 2017*; *Angermueller et al., 2017*; *Alley et al., 2019*). Compared to other deep learning approaches, RNNs are unique in that they are designed to handle variable length sequences, which makes them well-suited for applications involving proteins. Using only raw protein sequences as input, RNNs can learn the relevant positional dependencies of amino acids needed to associate each sequence with a corresponding functional value or values. Through this architecture, PARROT is able to capture intrinsic patterns in large protein datasets in order to construct highly accurate predictive models.

In this paper, we introduce the underlying RNN architecture of PARROT and demonstrate its application to three different biological problems. First, we show that PARROT performs at a near state-of-the-art level on phosphorylation site prediction tasks, a well-characterized bioinformatics problem. Second, we use PARROT to train a predictor of transcriptional activation activity using the extensive peptide library from *Erijman et al., 2020*. Third, we demonstrate how PARROT can be used in conjunction with DMS assays, using the amyloid beta-based dataset from *Seuma et al., 2021*. Ultimately, we show that PARROT is an effective, generalizable, and easy-to-use machine learning tool that is applicable to a range of different protein datasets.

## Results

### PARROT is a general RNN framework

Our motivation behind PARROT was to develop a powerful deep learning tool that is easy to implement into any large-scale protein analysis workflows (>1000s of sequences; *Figure 1A*). The general workflow involves the following steps. A user starts with a set of sequences of interest where each sequence (or each residue in each sequence) has some label associated with it, either a discrete class or a continuous value. PARROT uses this initial dataset to train, validate, and test a deep learning model. Training, validation, and testing are all performed automatically within PARROT using standard best practices for machine learning model generation. Once a model is built, the user can use that model to make predictions on new sequences for which there is no data associated.

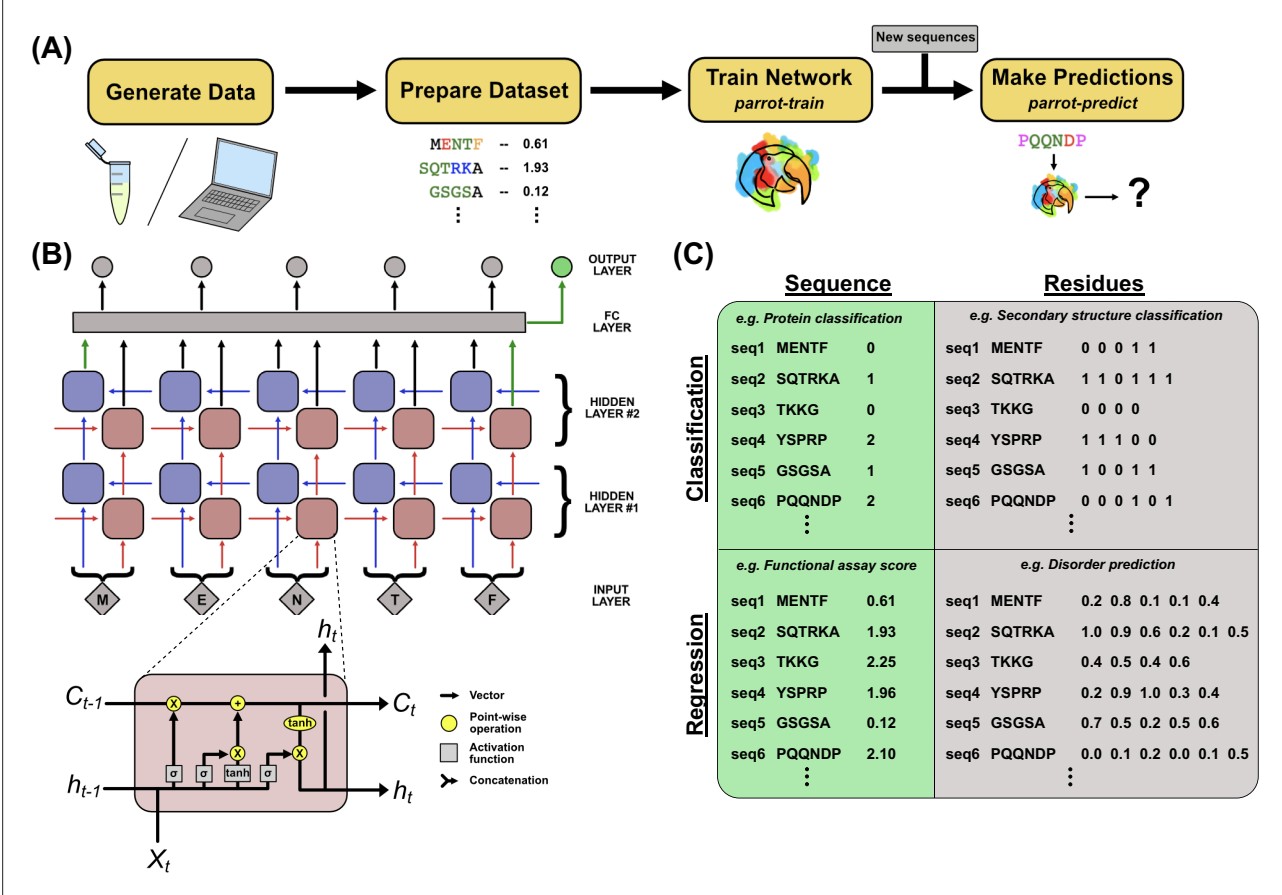

**Figure 1.** PARROT overview. (**A**) A standard workflow that incorporates PARROT. Quantitative protein data is either obtained computationally or generated through experiment, then formatted such that each protein sequence or residue is linked to a particular value. PARROT allows users to train a predictor on this dataset. The trained network can then be applied on new sequences to make predictions. (**B**) The internal architecture of PARROT is a bidirectional long short-term memory (LSTM) network. (Top) Series of cells propagate information along the length of a protein sequence in both N-to-C and C-to-N directions and the final output is integrated from the deepest layers in each direction. (Bottom) A diagram of the LSTM cells used in PARROT. (**C**) Example data formats for the four kinds of machine learning problems PARROT can carry out on proteins: classification or regression tasks using per-sequence or per-residue output.

We used the PyTorch platform to implement the core RNN framework of PARROT (*Paszke, 2021*). The serialized architecture of RNNs and their ability to handle variable length inputs makes them well-suited for learning information from protein sequences. In the context of protein analysis, each cell in an RNN integrates information from a particular amino acid with the output ('hidden state vector') of the preceding cell in the network. However, there are two main drawbacks of using the standard RNN architecture for protein analysis. First, standard RNNs require that information is propagated through the network in a single direction, which imposes an arbitrary constraint on the ability of a network to learn from protein sequences. Second, standard RNNs are susceptible to the 'vanishing gradient problem,' which arises due to the multiplication of many small values and can limit the ability of a network to learn long-range dependencies in the data (*Bengio et al., 1994*). PARROT implements two common variants of RNN architecture in order to mitigate these factors (*Figure 1B*). To address the first, the RNN implementation of PARROT is *bidirectional*, such that there are two parallel RNNs that propagate information from the protein sequence in opposite directions (N-to-C and C-to-N) simultaneously (*Schuster and Paliwal, 1997*). To address the issue of vanishing gradients, PARROT employs long short-term memory (LSTM) cells, which have been shown to capture long-range dependencies in sequences more efficiently than standard RNNs (*Hochreiter and Schmidhuber, 1997*). Combining bidirectionality with LSTM cells has been previously demonstrated to be effective at learning from biological sequences (*Hanson et al., 2017*; *Heffernan et al., 2017*; *Almagro Armenteros et al.,*

*2017*; *Li et al., 2017*; *Alley et al., 2019*). Taken together, PARROT's underlying network architecture is specifically optimized for working with protein datasets.

PARROT was designed to conceal the inner workings of this RNN, such that only a limited set of information is required from the user. For the most basic usage, the user only needs to provide their data and specify the kind of machine learning task for which they are training the network (classification or regression, described below). User datasets are input as basic whitespace-delimited text files with each protein sequence and its corresponding data contained on a single line. This file can be prepared in any spreadsheet program (e.g., Excel) and saved as a tab-separated variable file. More detailed instructions for file preparation are provided in the PARROT documentation. One of the consequences of PARROT's internal RNN is that the provided input sequences are not required to be the same length. Before training a PARROT network, users must specify whether their application qualifies as a *classification* or *regression* task. In classification tasks, the network is trained to assign discrete class labels to each input. For example, if one had a set of proteins where each protein localized to a specific organelle, this would lend itself to a classification task for predicting subcellular localization. For regression, the network outputs a continuous, real-number value for each input. For example, if one had a set of peptides where each sequence had an aggregation score between 0 and 1, this would lend itself to a regression task for predicting quantitative peptide aggregation. In addition to these two categories, users must also specify whether they want the PARROT network to produce *per-sequence* or *per-residue* output. Example data formats for each of these four categories are depicted in *Figure 1C*. Beyond this core usage, advanced users may optionally specify network hyperparameters such as the number of layers in the network, size of the hidden state vectors, learning rate, batch size, number of training epochs, and various other optional arguments (see Materials and methods).

In the remaining sections, we demonstrate the effectiveness of PARROT in the context of three distinct protein applications. Our goal here is to illustrate the diverse types of biological questions PARROT is capable of interrogating and to inspire readers to apply PARROT in their own research.

## PARROT predicts phosphosites on par with established methods

We first benchmarked the performance of PARROT-derived networks on a commonly studied bioinformatics task: predicting phosphorylation sites in a protein sequence. We used the Phospho. ELM (P.ELM) version 9.0 (*Diella et al., 2007*) and PhosPhAt (PPA) version 3.0 (*Heazlewood et al., 2008*; *Durek et al., 2010*) datasets for training and independent validation, similar to *Dou et al., 2014*. P.ELM consists of literature-derived animal phosphorylation sites, and PPA consists of mass spectrometry-validated phosphosites in *Arabidopsis thaliana*. For both of these datasets, we extracted all 19-residue windows centered around serine, threonine, and tyrosine and divided each of these into phosphorylation-positive and -negative sets. After filtering out similar sequences with CD-HIT (*Fu et al., 2012*), we then downsampled the larger phosphorylation-negative sets in order to create balanced datasets with identical numbers of phosphorylation-positive and phosphorylation-negative windows. Separate PARROT networks were trained on the serine, threonine, and tyrosine windows from the P.ELM dataset (*Figure 2A*).

We first tested our PARROT phosphosite predictors for each of the three residues on the P.ELM dataset using 10-fold cross-validation. This involved randomly splitting each residue-specific dataset into 10 even subsets, then training on 9/10 of the data and testing on the held out 1/10 for each of the subsets. As a benchmark, we compared the performance of our PARROT networks against three established phosphosite predictors, PhosphoSVM, MusiteDeep, and PHOSFER, which each rely upon different methodologies (*Dou et al., 2014*; *Trost and Kusalik, 2013*; *Wang et al., 2017*). As this was a binary classification problem, we focused our analysis on sensitivity, specificity, and Matthew's correlation coefficient (MCC) as performance metrics. We chose MCC as a performance metric because it has been shown to be more informative for binary classification tasks than the more commonly used F1 score or accuracy (*Chicco and Jurman, 2020*). Overall, the PARROT networks performed better than PhosphoSVM, and at a comparable level to PHOSFER and MusiteDeep (*Figure 2B*, *Supplementary file 1*). Interestingly, there was variation in the relative performance of the top three methods across the three residue types, with PARROT performing best on pSer, second best on pThr, and third best on pTyr. This performance trend corresponds with the size of the training dataset available for each residue. The P.ELM cross-validation analysis also illuminated particular biases in each of the predictors.

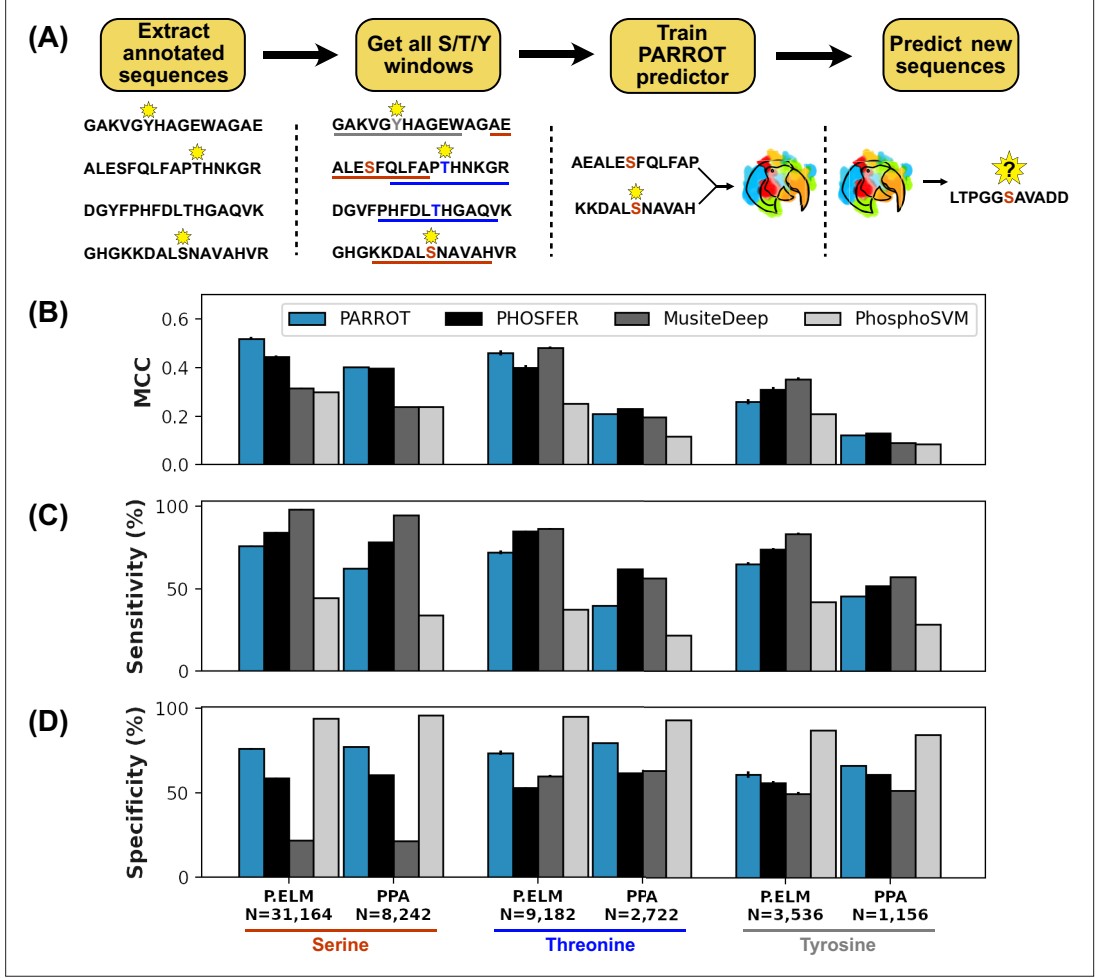

**Figure 2.** PARROT's performance on a phosphosite prediction task. (**A**) Workflow for training PARROT networks for phosphosite prediction. Full-length, annotated sequences from the Phospho.ELM (P.ELM) dataset were split into phospho-positive and phospho-negative 19aa windows (11aa windows used in figure for clarity). PARROT predictors trained on these sequence windows and were used to make predictions on held out sequences and the PhosPhAt (PPA) dataset. (**B**) Matthew's correlation coefficient (MCC), (**C**) sensitivity (**D**), and specificity scores for the PARROT predictors and three external predictors on the task of phosphosite prediction on the P.ELM and PPA datasets.

Notably, PHOSFER and MusiteDeep tended to predict with high sensitivity and low specificity, PhosphoSVM predicted with low sensitivity and high specificity (*Figure 2C and D*). However, PARROT's predictions tended to be the most balanced, with comparable sensitivity and specificity across the three different residue types.

Overfitting to training data is a common problem in the field of machine learning, so to test for this we assessed the performance of our PARROT predictors on an independent test dataset. For each of the three residue types, we trained a PARROT predictor on the full P.ELM dataset and made predictions on Ser, Thr, and Tyr residues in the PPA dataset. Unsurprisingly, all of the PARROT predictors performed worse on the PPA data than they did on the P.ELM cross-validation analysis; however, the PARROT predictors still performed comparably or better than the three established phosphorylation site predictors (*Figure 2B–D*, *Supplementary file 2*). PARROT's comparable performance to PHOSFER on the PPA dataset is particularly notable because PHOSFER was specifically designed for the prediction of plant phosphorylation sites (*Trost and Kusalik, 2013*).

Ultimately, our intention behind these analyses is not to assert that our PARROT-based predictors are inherently superior to all other existing phosphorylation site predictors. Rather it is to demonstrate that PARROT, despite being a general framework for any type of protein analysis, can perform equivalently to methods that are specifically developed for a particular task. In doing so, we establish that

PARROT-trained networks perform at a high level and that PARROT can confidently be extended to other less well-characterized protein applications.

## PARROT can integrate into high-throughput experiment workflows

Having established that predictors trained with PARROT can accurately learn patterns in large datasets, we next focused on showcasing PARROT's ability to integrate into a typical high-throughput experiment workflow. To accomplish this, we turned to the data generated by *Erijman et al., 2020*, in which the authors developed a high-throughput fluorescence assay for testing 30-mer peptides for activation domain (AD) function in yeast. Their assay measured ~37,000 sequences with AD function and ~1 million without, allowing them to train a convolutional neural network to predict AD function from sequence and secondary structure information (*ADpred*). This general workflow of (1) generating massive quantities of data using a high-throughput assay and (2) developing a computational predictor based on the assay data is becoming increasingly common in molecular biology. While ultimately the approach taken by Erijman et al. was computationally rigorous and successful, here we demonstrate that PARROT could readily be implemented in such a workflow without sacrificing performance (*Figure 3A*). Using PARROT in cases like this could save researchers valuable time from having to develop their own machine learning predictors from scratch.

Using 10-fold cross-validation, we trained PARROT networks on this AD dataset (see Materials and methods) and evaluated their performance and generalizability. First, we tested how well each of the networks predicted AD function on the held-out test set. PARROT networks predicted AD function with an accuracy of 93.1% (standard error ±0.1%) and an area under the precision-recall curve (AUPRC) of 0.973 (± 0.001), which were not significantly different from *ADpred*'s reported accuracy and AUPRC of 93.2% (± 0.1%) and 0.975 (± 0.001), respectively (*Figure 3B*). However, the PARROT-based predictors did significantly outperform the simple logistic regression model also used in *Erijman et al., 2020*, which had an accuracy of 89.1% (± 0.4%) and AUPRC of 0.942 (± 0.002). We also assessed the generalizability of the PARROT predictors through a similar approach as in the *ADpred* paper. Each cross-validation-trained network was also applied to an independent yeast AD dataset from *Staller et al., 2018*. We found good correlation between our predicted AD values and the independent data with an average Pearson's $R = 0.586$ (± 0.005), which was slightly higher than the reported performance of *ADpred* of $R = 0.57$ (*Figure 3C*).

To assess how the PARROT networks performed with fewer sequences to train on, we repeated both of these analyses on reduced datasets. Sampling from the complete dataset containing 75,846 30-mer peptides (50% displaying AD function), we created new 70K, 60K, 50K, 40K, 30K, 20K, 10K, and 5K peptide datasets. AUPRC began to plateau around 40K peptides, and generalizability to the Staller et al. data plateaued at around 30K, indicating that PARROT can robustly capture meaningful patterns in reduced datasets (*Figure 3D*).

Although all of the peptides studied in this analysis were 30 residues in length, one of the benefits of PARROT over other deep learning approaches is that it is not limited to fixed length sequences. In theory, one could train a predictor with PARROT using the combined results from multiple independent assays that test for similar phenotypes. As a proof of concept for this idea, we combined the data from Erijman et al. with the results from a similar AD functional assay that tested 5–20 residue peptides (*Ravarani et al., 2018*), trained new PARROT predictors on a variety of dataset sizes, and repeated the analyses described above. We found that 10-fold cross-validation accuracy and AUPRC slightly decreased using the combined datasets, possibly due to greater intra-dataset variance. However, performance on the independent test dataset was not significantly different (*Figure 3—figure supplement 1*). Despite the modest dip in performance for this particular case, we posit that PARROT's flexibility to incorporate multiple datasets while training could be useful in other contexts where a single, comprehensive dataset is not available.

As a final set of analyses, we compared our PARROT predictor to a recently published deep learning-based method for activation domain prediction, called *PADDLE,* developed by *Sanborn et al., 2021*. Similar to *ADpred*, *PADDLE* is a deep convolutional neural network and was trained on data derived from a quantitative, high-throughput assay. When applying our PARROT predictor trained on the Erijman et al. data to the Sanborn et al. data., we obtained relatively poor predictive power (*Figure 3—figure supplement 2*). However, since *ADpred* had also been shown to be ineffective at predicting the data obtained by *Sanborn et al., 2021*, we suspected that PARROT's

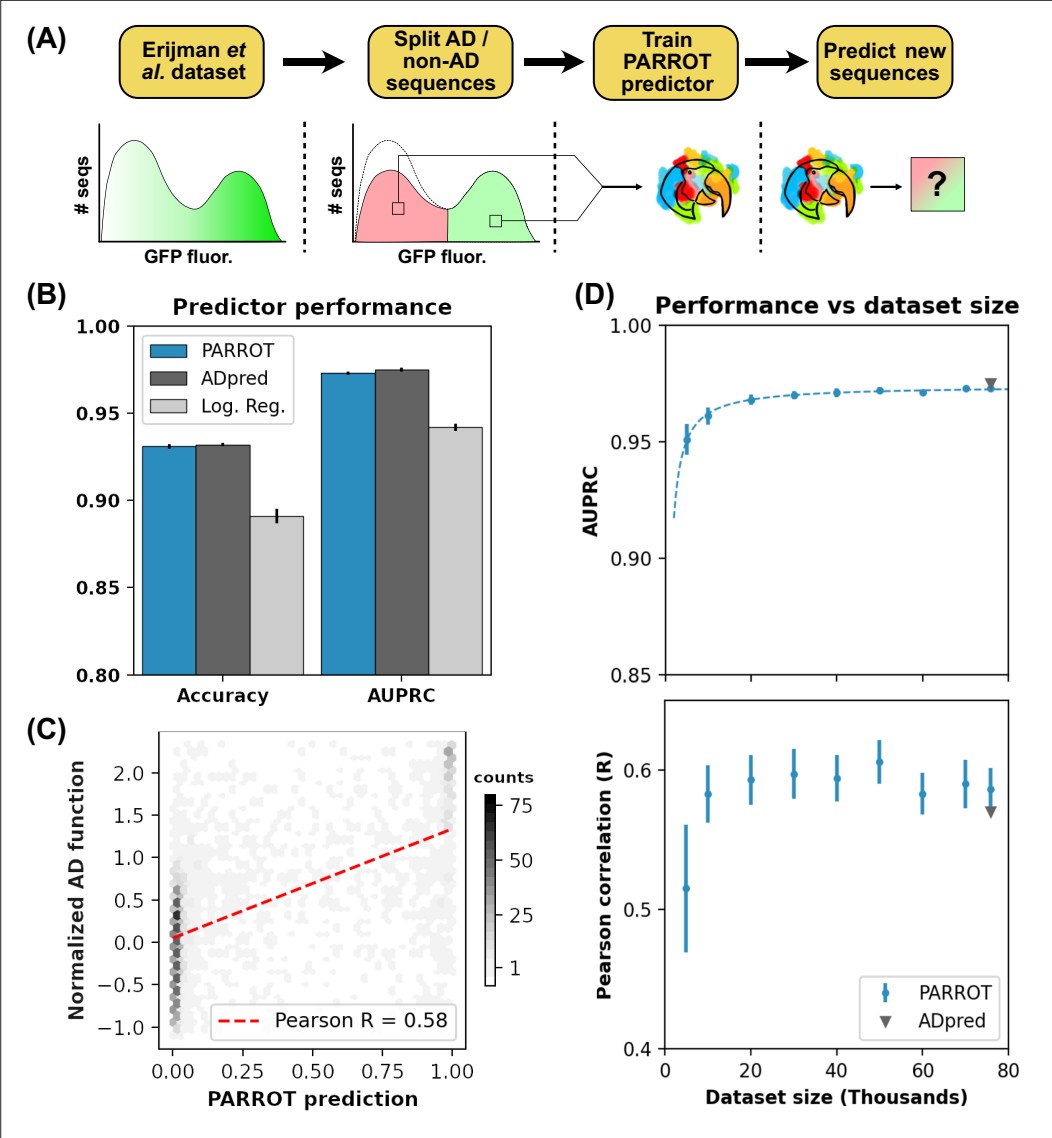

**Figure 3.** PARROT predicts functional yeast activation domains. (**A**) Diagram of activation domain workflow. A PARROT network was trained on the yeast fluorescence activation assay data from Erijman et al. and used to make predictions on new protein sequences. (**B**) PARROT's 10-fold cross-validation accuracy and area under the precision-recall curve (AUPRC) on the Erijman et al. dataset compared to the reported scores for two approaches employed in that paper: *ADpred* and a logistic regression based method. (**C**) Representative example of the correlation between PARROT's predictions and the true activation scores of an independent yeast activation domain dataset. (**D**) PARROT's performance on the tasks in (**B**) (top) and (**C**) (bottom) as a function of dataset size. For each specified dataset size, the actual number of sequences used for training and validation was 90% of the indicated value since networks were trained using 10-fold cross-validation. The dashed line is a hyperbola best-fit line. The reported performance of *ADpred* is shown for reference in gray.

The online version of this article includes the following figure supplement(s) for figure 3:

**Figure supplement 1.** Performance of PARROT networks trained on a multi-study dataset on the activation domain prediction task.

**Figure supplement 2.** Analysis of PARROT networks on the test set data of Sanborn et al.

underperformance may reflect inherent system-specific limitations in transferability between the two datasets. To test this, we leveraged PARROT's flexibility and trained a new predictor using the same training data as *PADDLE*. This new predictor saw substantially improved performance and was able to predict activation domain function comparably to *PADDLE*.

## PARROT can complement DMS experiments

For our final analysis, we demonstrate a unique usage for PARROT in tandem with DMS experiments. We conducted our training and testing of PARROT networks using a recent DMS dataset investigating amyloid beta (Aß42), a 42-residue peptide that can form plaques implicated in Alzheimer's disease (*Seuma et al., 2021*; *Findeis, 2007*). In work by Seuma et al., the authors tested >450 single and >14,000 double mutants of Aß42 in an assay that measured each variant's propensity to nucleate amyloid fibrils. Each of the variants they tested was assigned a log-ratio score (normalized to WT) with positive values indicating that that variant was nucleation-prone. While this scale of this experiment was massive, the sheer combinatorics of DMS makes it infeasible to truly capture all possible single and double mutations for a peptide of this size in a single assay. In our analysis, we show that PARROT can be employed to 'fill in the gaps' of a DMS experiment by training on the experimental variants and applying the network to predict the experimental outcome for variants that were not directly assayed.

We first validated PARROT's ability to predict nucleation scores from DMS data. Unlike the previous applications, the peptides obtained from DMS experiments occupy a relatively limited region of sequence space given that each sequence differs by only a few point mutations. It was not initially clear to us if PARROT would be able to learn the general, underlying patterns within this more focused dataset rather than overfitting on specific observations. To test this, we set out to rigorously evaluate our PARROT networks by developing and applying a method of *residue-wise cross-validation*. For each cross-validation fold, the held-out test set consisted of the data of all variants (single and double) linked to a particular residue in the sequence, while the training set consisted of all other variants. For example, during the round of cross-validation for the fourth position of Aß42 (Phe-4), variants like F4G, F4S, F4S-H13N, etc., would be excluded from the training data and held in the test set. While this approach to training might seem excessive, it avoids the issue of overfitting that would arise using conventional cross-validation training. For example, if we were to naively divide our DMS data into 10 random subsets, we could have cases where the training set consists of double mutants like F4S-H13N and F4S-V36M while the single mutant F4S is in the held-out test set. In this kind of situation, our predictions would be more accurate, but this would be improperly overestimating PARROT's performance.

Using residue-wise cross-validation, we trained and tested PARROT networks for all 42 positions of Aß42, taking the average predictions of double mutants since they were represented in the two separate test sets. Across all of the single and double mutants in the dataset, we see good correlation between PARROT's predictions and the true assay scores ($R^2$ = 0.593; *Figure 4B*). To provide context for this value, between multiple biological replicates of the DMS experiments an $R^2$ of 0.72 was obtained, indicating to us that PARROT is effectively capturing much of the variation between sequences that are not due to biological noise (*Seuma et al., 2021*). Within our entire set of predictions, the correlation was tighter among the double mutants in the dataset than the single mutants, likely due to the limited information that PARROT sees for the single mutants during training (*Figure 4—figure supplement 1*).

We next sought to see if the PARROT networks could capture epistatic relationships between Aß42 residues in the set of double mutants. In assays that measure complex phenotypes such as the nucleation of amyloid fibrils, it is not clear a priori if independent mutations will work synergistically or antagonistically when combined. For this analysis, we were interested in how well PARROT could predict the impact of double mutations in the DMS dataset relative to simpler estimations, such as summing the assay score of the two single mutations. Looking at only the double mutants in our dataset for which both point mutations were represented in the set of single mutants, we found that PARROT's predictions significantly outperformed this simple summing approach (p<0.01; *Figure 4C*). We also tested PARROT against other approaches for predicting double mutants: averaging the single mutant scores, taking the minimum score, or taking the maximum score, and similarly found that PARROT's predictions had significantly tighter correlation to the true values (*Figure 4—figure supplement 2*). While the effect size was relatively small, it is important to note that the PARROT networks making these epistatic predictions are training without key positional information due to the residue-wise cross-validation process. PARROT is not simply integrating information from the two single mutants, but rather it is making predictions based on general patterns it has learned from other variants.

Lastly, we wanted to see if PARROT was an effective tool for prioritizing untested candidate variants for follow-up study. Since it is infeasible for DMS experiments to test all possible point mutations

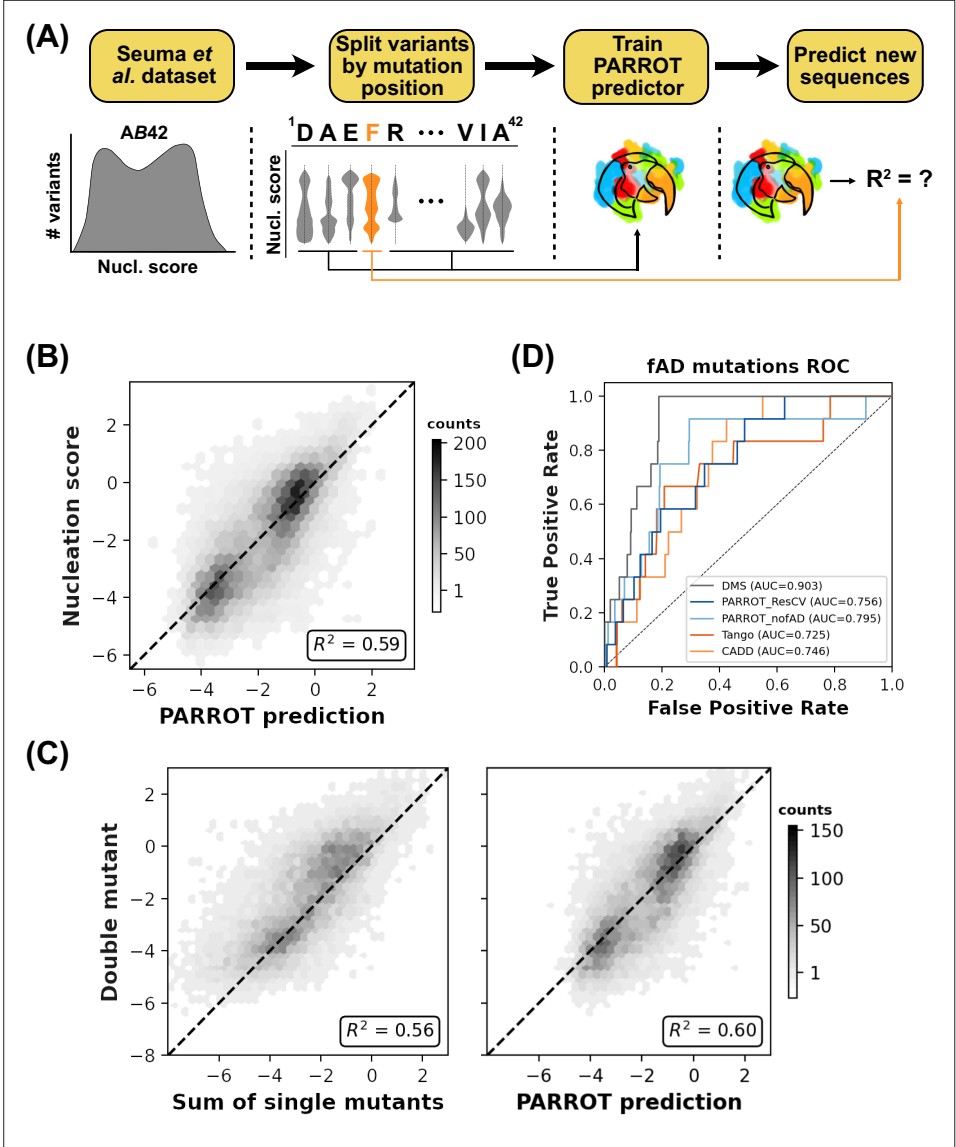

**Figure 4.** PARROT can 'fill in the gaps' of deep mutational scanning experiments. (**A**) Depiction of the residue-wise cross-validation workflow for predicting fibril nucleation scores using the Aβ42 deep mutational scanning (DMS) assay from Seuma et al. (**B**) Correlation between the true assay scores and predictions made by PARROT networks trained using residue-wise cross-validation for >14,000 single and double mutant variants. (**C**) Measurement of epistasis within the nucleation assay. (Left) Correlation between the nucleation scores of double mutants and the sum of nucleation scores of their composite single mutants. (Right) Correlation between the same double mutant nucleation scores and the predictions made by PARROT. (**D**) Receiver operator characteristic (ROC) curves for 12 familial Alzheimer's disease (fAD) mutants versus all other single mutant variants in the dataset. Area under the curve (AUC) values are reported in the legend.

The online version of this article includes the following figure supplement(s) for figure 4:

**Figure supplement 1.** Related to *Figure 4B*, correlation between Aβ42 nucleation scores and PARROT predictions divided into (**A**) single mutants and (**B**) double mutants.

**Figure supplement 2.** Related to *Figure 4C*, measured epistasis between Aβ42 double mutant nucleation scores and the average (top), maximum (middle), and minimum (bottom) of their composite single mutant scores.

in the protein sequence, we reasoned that PARROT might be an effective tool for making predictions on the mutants not covered by the assay. To test this idea, we assessed how effectively PARROT prioritized a set of 12 Aβ42 variants linked to familial Alzheimer's disease (fAD) within the entire collection of single mutants. This analysis was analogous to what was performed by Seuma and colleagues in

the original DMS study (*Seuma et al., 2021*). In addition to the predictions made by our residue-wise cross-validation networks (*PARROT_ResCV*), we trained an additional network using PARROT on the *entire* DMS dataset minus the 12 fAD-linked single mutants and all double mutants containing one or both of these mutations (*PARROT_nofAD*). We calculated area under the ROC curve (AUROC) to evaluate the predictions of these PARROT networks and compared PARROT's performance to the original DMS assay and to TANGO (*Fernandez-Escamilla et al., 2004*) and CADD (*Rentzsch et al., 2019*), which are computational predictors of aggregation and variant effect, respectively (*Figure 4D*). With the exception of the assay's scores (which PARROT trained on), *PARROT_nofAD* and *PARROT_ResCV* outperformed all other predictors. In particular, the success of the *PARROT_nofAD* predictor demonstrates that PARROT can effectively 'fill in the gaps' of DMS experiments and help prioritize candidate variants for follow-up study. Essentially, researchers can use PARROT to construct their own variant effect predictor that is specific to their assay and protein of interest.

## Discussion

When designing PARROT, we set out to develop a machine learning tool that effectively extracts patterns from protein sequence data, is generalizable to a wide array of regression and classification tasks, and is easy to use. There are a number of tools in recent years that satisfy some of these criteria, but not all three. For instance, deep learning-based predictors are becoming widely used in protein analysis, but these implementations tend to be designed for a single specific application rather than general use (*Heffernan et al., 2017*; *Almagro Armenteros et al., 2017*; *Alipanahi et al., 2015*). Although general protein analysis tools do exist, these typically implement simpler techniques like linear or logistic regression, support vector machines or decision trees, and are not necessarily able to identify complex, nonlinear patterns in datasets (*Brandes et al., 2016*; *Liu, 2019*). Meanwhile, open-source software packages like PyTorch, Keras, and TensorFlow make general deep learning frameworks freely available, but implementing these requires significant computational expertise and time investment. PARROT offers a freely available deep learning tool that satisfies all three of these criteria. By creating a tool that is sufficiently flexible, straightforward, and computationally rigorous, we aim to make the advantages of deep learning accessible to all biologists.

Importantly, we have demonstrated that predictors built using PARROT perform comparably to existing machine learning predictors across multiple contexts. In the case of phosphorylation site prediction, PHOSFER, PhosphoSVM, and MusiteDeep have all been specifically designed for this task, while PARROT was not. Nonetheless, PARROT still predicts phosphorylation sites approximately equivalently to each of these methods. Likewise, PARROT also performs comparably to both *ADpred* and *PADDLE* after training on the same dataset as either of these predictors. In our analysis of Aß42, we saw that PARROT networks trained on the DMS dataset were more effective at identifying pathogenic, fibril forming variants than computational tools like TANGO or CADD. Collectively, these results demonstrate that PARROT's flexibility across datasets does not come at the expense of performance. Moreover, while there has been a previous focus on the application of deep learning to understand folded protein stability, PARROT is demonstrably well-suited for working with intrinsically disordered protein sequences (*Alley et al., 2019*; *Cao et al., 2019*; *Hoie et al., 2021*; *Lindorff-Larsen and Kragelund, 2021*).

The three specific applications we used to showcase PARROT outline broader use cases in which it can be effective. For starters, PARROT can be used to create predictors from existing bioinformatic datasets; for example, we trained networks to predict phosphosites using the existing P.ELM dataset. Second, PARROT can easily be incorporated into the workflows of high-throughput protein experiments, as shown with the yeast activation domain predictor we created from Erijman et al.'s fluorescence assay data. DMS experiments are a special subset of this kind of usage. Our third example demonstrated how PARROT can train on DMS data and extrapolate predictions on variants that were not experimentally tested. In all three cases, PARROT can save researchers valuable time by eliminating the need to develop machine learning predictors de novo.

Beyond these applications, there are several other features built into PARROT that may increase its appeal to a wider scientific audience. Trained PARROT networks are fully portable into Python, which allows them to be easily integrated into stand-alone software tools, entirely independent of PARROT. As an example, we recently used PARROT to train a predictor of per-residue intrinsic disorder or predicted structure that offers a number of advantages in terms of performance and ease of use

compared to the state of the art (*Emenecker et al., 2021*). Additionally, while PARROT uses one-hot encoding to transform amino acid sequences into machine-readable numeric vectors by default, it can readily adopt other user-specified encoding schemes such as describing amino acids by their biophysical properties. As a consequence of this fact, PARROT is not specific to the canonical amino acid alphabet and can even be applied to nucleotide sequences. All of these features, and much more, are described in detail in the PARROT documentation.

As a final point, we would like to emphasize to prospective users of PARROT, or any similar tool, that predictions made by machine learning models should be interpreted with caution. Although deep learning methods are powerful at detecting patterns in data, this power also comes with increased susceptibility to overfitting and biased datasets. Proper data processing, not specific model architecture, is arguably the most critical factor for ensuring that deep learning is utilized accurately and meaningfully. While deep learning-based predictions can be instrumental in generating follow-up candidates and developing hypotheses, it is important to remember that these predictions do not replace the need for direct experimental validation.

## Materials and methods
### LSTM implementation
PARROT's underlying bidirectional LSTM network is implemented using the PyTorch library in Python. Input protein sequences are converted to one-hot vectors and grouped into batches (default: 32 sequences per batch), then fed into both the first forward layer and first reverse layer of LSTM cells. By default, PARROT networks consist of two layers of LSTM cells, though this hyperparameter can be manually specified by the user. Information is propagated between adjacent LSTM cells and between layers through hidden state vectors, which can also have a manually specified size (default 10). Hidden state vectors from the final layer of LSTM cells are converted to the final output via a fully connected linear or softmax neuron (*Figure 1C*). PARROT uses either a many-to-one or many-to-many architecture depending on whether the machine learning task at hand involves mapping protein sequences to single values (or class labels) or mapping *each residue* to a value/class label. The key implementation difference between these two architectures is in which hidden state vectors of the final layer of LSTM cells are input into the fully connected layer. For residue mapping, the hidden state vectors of the final forward and reverse cells *at each position in the sequence* are integrated into their own final connected layer (*Figure 1C*, gray). In contrast for sequence mapping, only the hidden state vectors from the final forward and final reverse cells are integrated into the fully connected layer (*Figure 1C*, green). For classification tasks, the fully connected layer outputs a vector with a size corresponding to the number of class labels. For regression tasks, this layer outputs a single value.

During training, weights in PARROT networks are updated using the Adam optimizer (*Kingma and Ba, 2014*). By default, the initial learning rate is set at 0.001. Classification tasks employ a cross-entropy loss function, while regression tasks use L1 and L2 loss functions for sequence mapping and residue mapping tasks, respectively. PARROT splits input datasets 70-15-15 into training, validation, and testing datasets by default; however, these proportions can be manually specified via the '--set-fractions' argument. The validation set is not trained on, but used to assess network performance after each epoch of training. The test set is completely held out until after training has concluded in order to give an estimate for how generalizable the trained network is on unseen data. Approximate training times for different hyperparameters and dataset sizes are listed in *Supplementary file 3*. Further implementation details and information on additional run-time arguments can be found in the PARROT documentation.

### Evaluation metrics
In binary classification problems, each prediction falls into one of four cases: true positive (TP), false positive (FP), true negative (TN), and false negative (FN). We compared our PARROT networks to other predictors using a variety of performance metrics that describe distribution of predictions across each of these categories. These metrics are calculated in the following ways:

$$Accuracy = \frac{TP + FP}{TP + FP + TN + FN} \tag{1}$$

$$Sensitivity \ = \ \frac{TP}{TP + FN} \tag{2}$$

$$Specificity \ = \ \frac{TN}{TN + FP} \tag{3}$$

$$Precision \ = \ \frac{TP}{TP + FP} \tag{4}$$

$$F1 \ Score \ = \ \frac{TP}{TP + 0.5 * (FN + FP)} \tag{5}$$

$$MCC \ = \ \frac{TP * TN - FP * FN}{\sqrt{(TP + FP)(TP + FN)(TN + FP)(TN + FN)}} \tag{6}$$

Alternatively, performance on classification tasks can be evaluated using precision-recall or receiver operator characteristic (ROC) curves. Instead of assigning each predicted sequence a discrete class label, sequences are assigned a continuous real number value corresponding to the confidence that it belongs to a particular class. We generated these non-discrete predictions using the optional '--probabilistic-classification' command-line argument and calculated AUPRC and AUROC using the Python package scikit-learn (*Pedregosa et al., 2011*).

## Phosphosite prediction

The same P.ELM and PPA datasets were used as by *Dou et al., 2014*, each split into separate phospho-serine, -threonine, and -tyrosine subsets. Initially, sequences with >30% similarity within each subset were removed using CD-HIT with default arguments (*Fu et al., 2012*). We next extracted all 19-residue windows centered around all serine, threonine, and tyrosine residues in each of the respective datasets, dividing these into phosphorylation-positive and phosphorylation-negative sets. A subsequent round of filtering was performed and sequences within these subsets with >20% similarity were removed. We then randomly downsampled the phosphorylation-negative sequences so that their number equaled the phosphorylation-positives and merged the two datasets into a single file for training by PARROT.

Our analysis proceeded by training and evaluating the networks on the P.ELM dataset using 10-fold cross-validation. The pSer, pThr, and pTyr datasets were each split randomly into 10 equal subsets. The PARROT script *parrot-cvsplit* facilitates this process of splitting a dataset into cross-validation subsets. Using the '--split' flag, PARROT networks were subsequently trained on nine of these sets and the resulting network made predictions for the sequences in the held out test set. These networks were trained using the following arguments: two hidden layers; hidden vector size of 10; learning rate of 0.0001; batch size of 64; 500 training epochs. The reported performance metrics in *Figure 2* and *Supplementary files 1 and 2* denote the average scores across the 10 cross-validation test sets. Predictions were also made by PHOSFER and MusiteDeep through their online web server on each of the cross-validation test sets and performance metrics were averaged. However, we opted not to test PhosphoSVM in this manner since this predictor was originally trained on the same P.ELM data and we wanted to avoid overfitting. Instead, we report the performance metrics taken directly from Dou et al. since these were calculated using a similar strategy of 10-fold cross-validation on the P.ELM dataset (*Dou et al., 2014*).

Using the same training arguments, additional networks were trained on the full P.ELM dataset (separately for pSer, pThr, and pTyr) and used to make predictions on the PPA dataset. Predictions were also made by PHOSFER, MusiteDeep on the same PPA data, and performance metrics were calculated for each of these sets of predictions. As with the P.ELM data, the performance metrics of PhosphoSVM on the PPA data were taken directly from Dou et al.

## Activation domain function prediction

The quantitative fluorescence assay data of Erijman et al. was collected and processed in a manner identical to its source paper (*Erijman et al., 2020*). Briefly, each 30-mer was assigned a real number score based on its distribution of reads across four fluorescence expression bins. These sequences were split into AD-positive and AD-negative sets and the negative set was sampled such that there were equal numbers of positive and negative sequences in the final dataset. This sampling process was repeated five times for the 'full' dataset (75,846 sequences), as well as for each of the reduced datasets (70K sequences, 60K sequences, etc.) in order to generate additional replicates.

Each dataset was split randomly into 10 cross-validation subsets, and PARROT networks were subsequently trained on nine and tested on the held-out subset. PARROT networks were trained

using the following hyperparameters: two hidden layers; hidden vector size of 10; learning rate of 0.0005; batch size of 64; 300 training epochs. Although our input data was set up as a classification task, by using the '--probabilistic-classification' argument, all of our predictions were output as real numbers between 0 and 1, which allowed us to conduct precision-recall curve analysis. In addition to assessing the performance on the held-out test set, each network was also used to make predictions on an independent dataset. This independent dataset was obtained from a similar yeast AD quantitative fluorescence assay from *Staller et al., 2018*. We calculated the normalized expression value for each sequence in this dataset by dividing the raw AD activity (GFP) by the protein expression level (mCherry), and log-normalizing the data around the WT sequence. The performance metrics reported in *Figure 3* are the averages of 50 total replicates (five replicate datasets with 10-fold cross-validation for each).

We also created a combined training dataset using the results from a similar AD functional assay in *Ravarani et al., 2018*. We extracted all sequences from this assay that were at least five residues in length and split into positive and negative sets as described using a cutoff of –0.14. These AD-positive and -negative sequences were then merged with the full Erijman et al. dataset, and PARROT networks were trained and evaluated in the same manner as before.

To perform comparisons against PADDLE (*Sanborn et al., 2021*), we extracted the activation assay data from Sanborn et al. and split into training and test sets as specified by the 'PADDLE split' column. A PARROT regressive model was trained on the full training set using the following hyperparameters: two hidden layers; hidden vector size of twenty; learning rate of 0.001; batch size of 64; 300 training epochs. Predictions were made on all of the test set sequences with this new network, as well as with the PARROT predictor that trained on the Erijman et al. data. Sequences in the test set that belonged to the transcription factor tiling, scramble mutant, and Pdr1 variant subsets were split and graphed separately.

## Aß42 nucleation prediction

Data linking Aß42 nucleation propensity to sequence was obtained from *Seuma et al., 2021*. Each single or double mutant variant was assigned a log-normalized (relative to WT) score with positive values reflecting that a variant is more prone to nucleating amyloid fibrils. For simplicity, we removed all nonsense variants from the dataset prior to training. The remaining variants were split into 42 different training-test set pairs, based on the position of the mutation(s) in that variant. Each test set contained all variants with mutations associated with a single residue, while the training sets consisted of all remaining variants. Accordingly, each double mutant was withheld in two separate test sets. Individual PARROT networks were trained on each of these unique training sets and the resulting network was used to make predictions on the corresponding test set. Networks were trained using the following hyperparameters: 3 hidden layers, hidden vector of size 8; learning rate of 0.0005; batch size of 64; and 250 training epochs. Predictions from the 42 test sets were combined, averaged (in the case of double mutants), and then analyzed.

We assessed the ability of PARROT to detect 'epistasis' by comparing the network's prediction of double mutants to simpler approaches that estimated mutant effect by integrating nucleation scores of the associative single mutations. We determined statistical significance between correlations derived from these different approaches through bootstrapping. All data points were resampled with replacement 10,000 times, calculating Pearson's $R$ for each iteration, and the 99% confidence intervals were used as a threshold for significance ($p < 0.01$).

The 12 fAD-linked variants that we analyzed were H6R, D7N, D7H, E11K, K16N, A21G, E22G, E22K, E22Q, D23N, L34V, and A42T. *PARROT_ResCV* and *PARROT_nofAD* predictions for all single mutants were ordered in order to create ROC curves. The CADD and TANGO predictions used for ROC analysis were also obtained from Seuma et al. as they performed an identical analysis on this set of 12 variants.

## Implementation

The complete PARROT implementation consists of four command-line commands: *parrot-train*, *parrot-predict, parrot-optimize and parrot-cvsplit*. For the analysis described here, *parrot-train* was used to train the RNN predictors given a properly formatted dataset and *parrot-predict* was used to make predictions on new sequences using an existing trained network. We did not use *parrot-optimize* in

these analyses, but can be used to automatically select network hyperparameters through Gaussian process optimization. *parrot-cvsplit* allows users to automatically split their datasets into k-folds for cross-validation. More details can be found in the PARROT documentation: https://idptools-parrot.readthedocs.io/. PARROT is optimized to run in a Mac or Linux environment, but can also work using Windows.

## Acknowledgements

We thank the members of the Holehouse lab for helpful discussions and feedback. Special thanks to Shubhanjali Minhas for designing the PARROT logo. Funding for this work was provided by the National Science Foundation grant number DGE-2139839 and the Longer Life Foundation (an RGA/Washington University collaboration).

## Additional information

### Competing interests

Alex S Holehouse: is a scientific consultant with Dewpoint Therapeutics. The other author declares that no competing interests exist.

### Funding

| Funder | Grant reference number | Author |
| --- | --- | --- |
| National Science Foundation | DGE-2139839 | Daniel Griffith |
| Longer Life Foundation | | Alex S Holehouse |

The funders had no role in study design, data collection and interpretation, or the decision to submit the work for publication.

### Author contributions

Daniel Griffith, Conceptualization, Data curation, Formal analysis, Investigation, Methodology, Software, Visualization, Writing – original draft, Writing – review and editing; Alex S Holehouse, Conceptualization, Funding acquisition, Investigation, Methodology, Software, Supervision, Writing – original draft, Writing – review and editing

### Author ORCIDs

Daniel Griffith (iD) http://orcid.org/0000-0002-9633-9601
Alex S Holehouse (iD) http://orcid.org/0000-0002-4155-5729

### Decision letter and Author response

Decision letter https://doi.org/10.7554/eLife.70576.sa1
Author response https://doi.org/10.7554/eLife.70576.sa2

## Additional files

### Supplementary files

• Supplementary file 1. Complete table of performance metrics for phosphosite predictions on the Phospho.ELM (P.ELM) dataset. Standard error, whenever possible, is reported in parentheses.

• Supplementary file 2. Complete table of performance metrics for phosphosite predictions on the PhosPhAt (PPA) datasets. Standard error, whenever possible, is reported in parentheses.

• Supplementary file 3. Average PARROT network training times on different sizes of datasets and with variable hyperparameters. Datasets were created by assigning random values in [–5, 5] to randomly generated protein sequences ~25–35 residues in length. Networks were trained using a NVIDIA TU116 GPU. Three replicate PARROT networks were trained for each specified set of hyperparameters and dataset.

• Transparent reporting form

## Data availability

All code is fully open source and available here: https://github.com/idptools/parrot. Documentation is available here: https://idptools-parrot.readthedocs.io/. Additional supporting data available here: https://github.com/holehouse-lab/supportingdata/tree/master/2021/griffith_parrot_2021 (copy archived at https://archive.softwareheritage.org/swh:1:rev:4bb48369891dc4416b6b176046846091 d8cd9ddb). PhosPhat was taken from http://phosphat.uni-hohenheim.de (specifically Phosphat_ 20200624.csv), while data for PhosphoElm where taken from http://phospho.elm.eu.org/. In both cases the entire dataset available at the time of analysis was used.

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
