## [Decision Letter]

**Acceptance summary:**

The analysis of large data sets obtained from omics or other approaches is often the most time consuming and difficult step of a study. Deep learning and related computational approaches offer the possibility to train a software on a certain data set and then analyze large new experimental data sets. The authors describe the software architecture and demonstrate the application of the system on three different topics: prediction of phosphorylation, prediction of transactivation potential of peptides and prediction of aggregation propensity. They compare the results of their new software PARROT with other existing software tools.

**Decision letter after peer review:**

Thank you for submitting your article "PARROT: a flexible recurrent neural network framework for analysis of large protein datasets" for consideration by *eLife*. Your article has been reviewed by 2 peer reviewers, including Volker Dötsch as the Senior and Reviewing Editor and Reviewer #2.

Essential revisions:

1) A suggestion would be to provide a bit more information about how non-experts can validate the results from PARROT, both in the documentation and main text. Because the authors are targeting non machine learning experts--and providing many useful defaults--they should think about how a non-expert might use the tool. Would a non-expert know when to use a ROC curve versus some other metric? Obviously, the authors should not write a ML textbook here, but think carefully through ways to guide users to appropriate tests.

Two ideas:

1) Have PARROT spit out a stack of test results by default, rather than the few it defaults to now. You could put this behavior under a flag (e.g. --testsoff) so more advanced users would not get bombarded by spew, but otherwise confront the user with as many validation metrics as possible.

2) Encode some warning heuristics for common errors. For example, PARROT could warn the naive users that their training set was unbalanced.

---

## [Author Response]

Essential revisions:1) A suggestion would be to provide a bit more information about how non-experts can validate the results from PARROT, both in the documentation and main text. Because the authors are targeting non machine learning experts--and providing many useful defaults--they should think about how a non-expert might use the tool. Would a non-expert know when to use a ROC curve versus some other metric? Obviously, the authors should not write a ML textbook here, but think carefully through ways to guide users to appropriate tests.Two ideas:1) Have PARROT spit out a stack of test results by default, rather than the few it defaults to now. You could put this behavior under a flag (e.g. --testsoff) so more advanced users would not get bombarded by spew, but otherwise confront the user with as many validation metrics as possible.2) Encode some warning heuristics for common errors. For example, PARROT could warn the naive users that their training set was unbalanced.

We thank the reviewers for their thoughtful and constructive suggestions. As PARROT is ultimately designed with the non-ML expert in mind, we agree that we could provide more features and information to help naive users. To address this, we made extensive changes that we hope will be helpful for all prospective users of PARROT. We also made several smaller, quality-of-life changes in the newest version of PARROT. The major changes we made include:

1. Streamlining k-fold cross-validation

Since cross-validation is one of the most widely-used forms of validation in machine learning (including in our analyses for this manuscript), we wanted to facilitate this process in PARROT and provide examples to show users how to carry it out themselves. In particular, we added the script *parrot-cvsplit* into the PARROT distribution. This short script will randomly and automatically divide a dataset into a user-specified number of folds for training/testing a network. To demonstrate this, we have provided a complete example of 10-fold cross-validation in the documentation under the page “Evaluating a Network with Cross-Validation”. We have also provided more details on the general principles of validating trained networks on the “Machine Learning Resources” page (under “How can I validate that my trained network is performing well?”)

2. Providing users with more performance stats

As the reviewers recommended, we also increased the amount of information that PARROT provides to users regarding the performance of their trained network on the test set data. Previous iterations of PARROT only provided a single file containing the true and predicted values for each sequence in the test set and optional performance figures. The newest version of PARROT provides this information along with an additional “performance stats” file that describes network performance on a variety of metrics. A brief description of each of these metrics, and links to more detailed resources, are provided on the “Machine Learning Resources” page (under “What are the different performance metrics for evaluating ML networks?”). We hope the combination of these metrics and cogent, simple explanations of their meaning will help users better assess their trained models.

3. Providing users with warning heuristics

Like the reviewers suggested, we also added a few different warnings that inform users if their dataset does not meet particular heuristics. These warnings check for class imbalance (if classification task), dataset skew/imbalance or non-standardized data (if regression task), duplicate sequences, and minibatch size compared to dataset size.

4. Documentation overhaul

In concert with all of the major and minor code changes, we completely overhauled PARROT’s documentation (https://idptools-parrot.readthedocs.io). The new docs reflect all of the additional features that have been added to PARROT. Additionally, we reworked the examples so that it is more organized and accessible. Now the examples are split into a “Basic Examples” and “Advanced Examples” page, and, as mentioned above, we added a full walkthrough of a cross-validation example.